# Relationship between Osteosarcopenia and Frailty in Patients with Chronic Liver Disease

**DOI:** 10.3390/jcm9082381

**Published:** 2020-07-26

**Authors:** Chisato Saeki, Tomoya Kanai, Masanori Nakano, Tsunekazu Oikawa, Yuichi Torisu, Masahiro Abo, Masayuki Saruta, Akihito Tsubota

**Affiliations:** 1Department of Internal Medicine, Division of Gastroenterology and Hepatology, The Jikei University School of Medicine, 3-25-8 Nishi-shimbashi, Minato-ku, Tokyo 105-8461, Japan; tomoyaaust@hotmail.com (T.K.); masanori-nakano@jikei.ac.jp (M.N.); oitsune@jikei.ac.jp (T.O.); torisu@jikei.ac.jp (Y.T.); m.saruta@jikei.ac.jp (M.S.); 2Department of Internal Medicine, Division of Gastroenterology, Fuji City General Hospital, 50 Takashima-cho, Fuji-shi, Shizuoka 417-8567, Japan; 3Department of Rehabilitation Medicine, The Jikei University School of Medicine, 3-25-8 Nishi-shimbashi, Minato-ku, Tokyo 105-8461, Japan; abo@jikei.ac.jp; 4Core Research Facilities, Research Center for Medical Science, The Jikei University School of Medicine, 3-25-8 Nishi-shimbashi, Minato-ku, Tokyo 105-8461, Japan

**Keywords:** chronic liver disease, osteosarcopenia, frailty, vertebral fracture

## Abstract

Osteosarcopenia and frailty have a negative health impact on an aging society. This cross-sectional study aimed to investigate the clinical characteristics and relationship of osteosarcopenia and frailty in 291 patients with chronic liver disease (CLD), who comprised 137 males and 154 females, with a median age of 70.0 years. Sarcopenia was diagnosed according to the Japan Society of Hepatology criteria. Bone mineral density was measured using dual-energy X-ray absorptiometry. Frailty was defined by five parameters (exhaustion, slowness, weakness, low physical activity, and weight loss). Among the 291 patients, 49 (16.8%) and 81 (27.8%) had osteosarcopenia and frailty, respectively. Frailty and vertebral fracture were more frequently noted in patients with osteosarcopenia than in those without osteosarcopenia (79.6% vs. 17.4% and 59.2% vs. 20.2%, respectively; *p* < 0.001 for both). Meanwhile, osteosarcopenia and vertebral fracture were more frequently observed in patients with frailty than in those without frailty (48.1% vs. 4.8% and 49.4% vs. 18.1%, respectively; *p* < 0.001 for both). On multivariate analysis, frailty was an independent factor associated with osteosarcopenia (odds ratio (OR), 9.837; *p* < 0.001), and vice versa (OR, 10.069; *p* < 0.001). Osteosarcopenia and frailty were prevalent, closely interrelated, and increased the risk of vertebral fracture in patients with CLD.

## 1. Introduction

Osteoporosis and sarcopenia are musculoskeletal disorders characterized by the loss of bone mass and skeletal muscle mass and strength, respectively [1,2,3,4,5]. Although both these disorders affect different organs and tissues, they are closely interrelated through common factors, such as genetic properties, endocrine hormones, nutrition conditions, and lifestyle behavior [2,6]. Accordingly, the concept and term “osteosarcopenia” have been established recently, which is defined as the concomitant occurrence of sarcopenia and osteoporosis [1,2,3,4]. Osteosarcopenia has a negative impact on health-related quality of life and eventual prognosis, with an increased risk of falls, fractures, institutionalization, and mortality [1,2,3,4]. Recently, we reported a relatively high prevalence of osteosarcopenia in 21.8% of patients with liver cirrhosis (LC), and a close association between osteosarcopenia and vertebral fractures [7]. Osteosarcopenia has now become the focus of attention in terms of health-related quality of life, especially in patients with chronic liver disease (CLD), including LC.

Frailty, defined as a state of multisystem physiological dysregulation and increased vulnerability to stressors, increases the risk of falls, disability, hospitalization, and mortality among the elderly and CLD patients, particularly those with advanced liver disease [8,9,10,11,12,13,14,15]. The prevalence of frailty in patients with advanced liver disease ranges from 17–49% [10,11,12,13,14,15], with frailty being associated with the presence of ascites, hepatic encephalopathy, and Child–Pugh class C [16]. Furthermore, among patients with end-stage liver disease requiring liver transplantation, those with frailty have significantly higher Model for End-Stage Liver Disease scores and mortality rates [11,12]. Therefore, early assessment and appropriate treatment of frailty are crucial to reduce the aforementioned risks, especially in patients with advanced liver disease.

Several geriatric studies have reported the relationship between osteosarcopenia and frailty in community-dwelling elderly individuals [17,18]. Among 1083 Japanese elders (≥60 years old), 5.6% developed frailty and 30.4% of the frail elders had osteosarcopenia, with the rate being higher in frail elders than in the non-frail and pre-frail elders [17]. Conversely, the occurrence of frailty increased, especially with the presence of osteosarcopenia. Similarly, among 316 Chinese community-dwelling elders (≥65 years old), 14.2% developed frailty and 33.3% of the frail elders had osteosarcopenia, with the rate being higher in frail elders than in the non-frail and pre-frail elders [18]. Conversely, the likelihood of an individual being frail or pre-frail was substantially higher with the presence of osteosarcopenia. Hence, there is a close relationship between frailty and osteosarcopenia in the community-dwelling elderly population. However, among patients with CLD, it was unclear whether similar relationships could be observed between frailty and other complications. In addition, it was uncertain how the prevalence of these complications in CLD patients differed from that in the community-dwelling elderly population.

This study aimed to investigate the prevalence of these complications, the relationship among them, and the factors associated with osteosarcopenia and frailty in patients with CLD.

## 2. Materials and Methods

### 2.1. Statement of Ethics

This study was conducted in accordance with the Declaration of Helsinki, and it was approved by the Ethics Committee of Fuji City General Hospital (approval No. 156).

### 2.2. Study Design and Patients

This was a cross-sectional study including 291 Japanese patients who were diagnosed with CLD at Fuji City General Hospital (Shizuoka, Japan) between 2017 and 2019. The inclusion criteria were as follows: (1) presence of CLD, such as chronic hepatitis B or C, alcoholic liver disease, autoimmune hepatitis, primary biliary cholangitis, and non-alcoholic steatohepatitis; (2) measurement of the skeletal muscle mass index (SMI) using bioelectrical impedance analysis (InBody S10; InBody, Seoul, Korea), grip strength using a dynamometer (T.K.K5401 GRIP-D; Takei Scientific Instruments, Niigata, Japan), and bone mineral density (BMD) using dual-energy X-ray absorptiometry (PRODIGY; GE Healthcare, Madison, WI, USA); (3) assessment of vertebral fractures using spinal lateral radiographs; (4) evaluation of frailty according to the criteria advocated by Fried et al. [19]. We excluded patients who had preexisting refractory ascites or implants, or had been undergoing hemodialysis, because the bioelectrical impedance analysis could overestimate SMI under these conditions [7].

### 2.3. Diagnosis of Osteoporosis, Sarcopenia, and Frailty

BMD was assessed at the lumbar spine (L2–L4), femoral neck, and total hip. Osteoporosis was diagnosed according to the World Health Organization criteria: T-score ≤ −2.5 for osteoporosis, between −2.5 and −1.0 for osteopenia, and > −1.0 for normality [20]. Sarcopenia was diagnosed according to the criteria proposed by the Japan Society of Hepatology [21]. Briefly, sarcopenia is defined as having low handgrip strength (<26 kg for males and <18 kg for females) and low muscle mass (SMI < 7.0 kg/m^2^ for males and <5.7 kg/m^2^ for females). Pre-sarcopenia and dynapenia were defined as having low muscle mass without low muscle strength and having low muscle strength without low muscle mass, respectively. SMI was calculated as the sum of the muscle mass of the four limbs divided by the height in square meters (kg/m^2^). Gait speed was assessed over a distance of 6 m, and low gait speed was defined as < 1.0 m/s. Frailty was diagnosed using a validated screening tool based on Fried’s five components [19,22]: weight loss (≥2 kg over the last 6 months); weakness (handgrip strength < 26 kg for males and < 18 kg for females); exhaustion (positive answer to the question: “In the last two weeks, have you felt tired without a reason?”); slowness (gait speed < 1.0 m/s); low physical activity (negative answer to the following two questions: “Do you engage in moderate levels of physical exercise or sports aimed at health?” and “Do you engage in low levels of physical exercise aimed at health?”). Absence of these five components was defined as non-frailty, presence of 1–2 components was defined as pre-frailty, and presence of ≥3 components defined as frailty [22].

### 2.4. Clinical and Laboratory Assessments

Blood samples were obtained from each patient after overnight fasting. The following serum parameters were measured using routine laboratory methods: total bilirubin, albumin, Mac-2-binding protein glycosylation isomer (M2BPGi; hepatic fibrosis marker), insulin-like growth factor-1 (IGF-1), zinc, branched-chain amino acids (BCAAs), tartrate-resistant acid phosphatase (TRACP)-5b (bone resorption marker), total procollagen type I N-terminal propeptide (P1NP; bone formation maker), intact parathyroid hormone (PTH), and prothrombin time-international normalized ratio (PT-INR).

### 2.5. Statistical Analysis

Continuous variables are presented as medians and interquartile ranges in parentheses. The Mann–Whitney U test was used to evaluate differences in the distribution of continuous variables between two groups, while the Kruskal–Wallis test followed by the Steel–Dwass post-hoc test was used for multiple comparisons among three or four groups. Categorical variables are presented as numbers and percentages in parentheses. Inter-group differences were evaluated using the chi-squared test. Univariate and multivariate logistic regression analyses were performed to identify variables that were significantly and independently associated with osteosarcopenia and frailty. Statistical analyses were performed using SPSS (version 26, IBM, Armonk, NY, USA), with a *p*-Value < 0.05 indicating statistical significance.

## 3. Results

### 3.1. Patient Characteristics

Baseline clinical characteristics of the 291 patients with CLD enrolled in the current study are shown in Table 1. This study cohort included 137 males (47.1%), with a median age of 70.0 (59.0–76.0) years. Among the 291 patients, 151 (51.9%) had LC. The median SMI and handgrip strength values were 6.48 (5.74–7.26) kg/m^2^ and 23.4 (17.9–31.4) kg, respectively. The median BMD values of the lumber spine, femoral neck, and total hip were 1.07 (0.90–1.22) g/cm^2^, 0.76 (0.67–0.89) g/cm^2^, and 0.83 (0.71–0.94) g/cm^2^, respectively. The prevalence of vertebral fracture was 26.8% (78/291).

### 3.2. Prevalence of Osteoporosis, Sarcopenia, and Frailty

The prevalence of sarcopenia, osteoporosis, and frailty among the 291 patients was 26.8% (78/291), 32.6% (95/291), and 27.8% (81/291), respectively (Figure 1A,B). Presence of all three complications was observed in 13.4% (39/291) patients. Presence of any two complications was noted as follows: 16.8% (49/291) with sarcopenia/osteoporosis (osteosarcopenia), 20.3% (59/291) with sarcopenia/frailty, and 17.9% (52/291) with osteoporosis/frailty. Presence of any one complication was observed as follows: 3.1% (9/291) with sarcopenia, 11.3% (33/291) with osteoporosis, and 3.1% (9/291) with frailty. Accordingly, 133 patients (45.7%) showed some complication, whereas 158 patients (54.3%) were free from these complications.

### 3.3. Comparison of Clinical Characteristics between Patients with and without Osteosarcopenia

Males accounted for 32.7% of the osteosarcopenia group and 50.0% of the non-osteosarcopenia group, with osteosarcopenia being more prevalent in females than in males (*p* = 0.027; Table 1). Osteosarcopenia patients were older (*p* < 0.001) and had a lower body mass index (BMI; *p* < 0.001) as compared to non-osteosarcopenia patients. Regarding biochemical parameters, the osteosarcopenia group had significantly lower levels of IGF-1 (*p* < 0.001) and BCAA (*p* < 0.001), and higher levels of M2BPGi (*p* = 0.030) and PTH (*p* = 0.010) as compared to the non-osteosarcopenia group. Notably, the osteosarcopenia group showed a significantly higher prevalence of frailty (79.6% vs. 17.4%), low gait speed (79.6% vs. 23.1%), and vertebral fracture (59.2% vs. 20.2%) than the non-osteosarcopenia group (*p* < 0.001 for all).

Next, the 291 patients were classified into the following four groups (Appendix A): (i) patients without both osteoporosis and sarcopenia (168/291; 57.5%); (ii) patients with osteoporosis alone (46/291; 15.8%); (iii) patients with sarcopenia alone (28/291; 9.6%); (iv) patients with both osteoporosis and sarcopenia (i.e., osteosarcopenia; 49/291; 16.8%). Among these four groups, the osteosarcopenia group showed the highest prevalence of frailty (79.6% (39/49); adjusted residual = |8.6|; Cramér’s V = 0.652), low gait speed (79.6% (39/49); adjusted residual = |7.7|; Cramér’s V = 0.554), and vertebral fracture (59.2% (29/49); adjusted residual = |5.6|; Cramér’s V = 0.371) (*p* <0.001 for all; Figure 2A–C).

### 3.4. Factors Associated with Osteosarcopenia in Patients with Chronic Liver Disease

The following eight variables showed a significant relation with osteosarcopenia on univariate analysis: gender, age, BMI, IGF-1, BCAA, PTH, frailty, and vertebral fractures (Appendix A). On multivariate analysis, the following five variables were retained as independent factors associated with osteosarcopenia (Table 2): lower BMI (odds ratio (OR), 0.821; 95% confidence interval (CI), 0.726–0.929; *p* = 0.002); lower levels of IGF-1 (OR, 0.980; 95% CI, 0.964–0.996; *p* = 0.014); higher levels of PTH (OR, 1.017; 95% CI, 1.005–1.030; *p* = 0.006); presence of frailty (OR, 9.837; 95% CI, 4.199–23.043; *p* < 0.001); vertebral fracture (OR, 3.306; 95% CI, 1.439–7.596; *p* = 0.005).

### 3.5. Comparison of Clinical Characteristics between Patients with and without Frailty

Males accounted for 38.3% of the frail group and 50.5% of the non-frail and pre-frail groups (*p* = 0.062; Table 3). Frail patients were older and had a lower BMI and higher prevalence of LC as compared to non-frail and pre-frail patients (*p* < 0.001 for all). The frail group had significantly lower levels of albumin (*p* = 0.001), IGF-1 (*p* < 0.001), zinc (*p* = 0.001), and BCAA (*p* < 0.001), and higher levels of M2BPGi (*p* < 0.001) as compared to the non-frail and pre-frail groups. The BMD values of the lumbar spine, femoral neck, and total hip were significantly lower in the frail group than in the non-frail and pre-frail groups (*p* < 0.001 for all). Notably, the frail group showed a significantly higher prevalence of osteosarcopenia (48.1% vs. 4.8%) and vertebral fractures (49.4% vs. 18.1%) as compared to the non-frail and pre-frail groups (*p* < 0.001 for both).

The 291 patients were classified into the following three groups (Appendix A): (i) patients without pre-frailty and frailty (non-frailty: 98/291; 33.7%); (ii) patients with pre-frailty (112/291; 38.5%); (iii) patients with frailty (81/291; 27.8%). The frail group had significantly lower SMI, handgrip strength, and BMD values than the non-frail and pre-frail groups (*p* < 0.001 for all; Figure 3A,B). Notably, the frail group had the highest prevalence of osteosarcopenia (48.1% (39/81); *p* < 0.001; adjusted residual = |8.9|; Cramér’s V = 0.526) and vertebral fracture (49.4% (40/81); *p* < 0.001; adjusted residual = |5.4|; Cramér’s V = 0.318) among the three groups (Figure 3C).

### 3.6. Factors Associated with Frailty in Patients with Chronic Liver Disease

The following 11 variables showed a significant association with frailty on univariate analysis: age, BMI, LC, albumin, M2BPGi, IGF-1, zinc, BCAA, PTH, osteosarcopenia, and vertebral fractures (Appendix A). On multivariate analysis, the following four variables were significantly and independently associated with frailty (Table 4): older age (OR, 1.090; 95% CI, 1.050–1.130; *p* < 0.001); higher levels of M2BPGi (OR, 1.149; 95% CI, 1.039–1.271; *p* = 0.007); lower levels of BCAA (OR, 0.994, 95% CI, 0.990–0.997; *p* = 0.001); presence of osteosarcopenia (OR, 10.069; 95% CI, 4.282–23.680; *p* < 0.001).

## 4. Discussion

Osteosarcopenia, a recently established syndrome, is defined by the coexistence of sarcopenia and osteoporosis. It has now become a global health concern, given that both of the disease conditions are risk factors for falls, disability, hospitalization, and mortality [1,2,3,4]. In the present study, the prevalence of osteosarcopenia was 16.8% among all patients with CLD and increased to 48.1% when limited to those with frailty (median, 76 years), whereas it was only 4.8% among those without frailty (median, 67 years). A community-based geriatric study of Chinese elders reported the prevalence of 33.3% of osteosarcopenia among frail elders (mean, 83–84 years) and 1.7% among non-frail elders (mean, 72–74 years) [18]. These findings indicate that individuals with frailty are more susceptible to osteosarcopenia, irrespective of the presence of CLD. Intriguingly, our study cohort was younger than the Chinese elderly cohort, and osteosarcopenia was more frequently noted in the former than in the latter; thus suggesting that CLD patients with frailty are more vulnerable to osteosarcopenia than the general population with frailty. In the present study, we found that frailty was most frequently noted in the osteosarcopenia group, and it was significantly and independently associated with osteosarcopenia. Therefore, early diagnosis and appropriate treatment for osteosarcopenia are required, especially in CLD patients with frailty.

Frailty is a complex syndrome characterized by a decline in multisystem functioning and physiologic reserve, combined with increased vulnerability to stressors, and an increase in the incidence of falls, disability, hospitalization, and mortality among elders and patients with advanced liver disease [8,9,10,11,12,13,14,15]. Despite advances in clinical research on frailty in patients with end-stage liver disease, the prevalence of frailty among those with CLD at different disease stages and its etiologies has not been fully elucidated. In the present study, we found that the prevalence of frailty was 27.8% among 291 patients with CLD, including four patients with end-stage liver disease. Meanwhile, the prevalence of frailty among community-dwelling elders was 5.6% in Japan and 14.2% in China [17,18]. These differences suggest that patients with CLD are more susceptible to frailty as compared to the general population. Our findings showed that (1) the frail group had significantly lower levels of SMI, handgrip strength, and BMD as compared to the non-frail and pre-frail groups; (2) osteosarcopenia occurred more frequently in the frail group as compared to the other groups; (3) frailty not only occurred more frequently in the osteosarcopenia group, but also showed a significant and independent association with osteosarcopenia; (4) conversely, osteosarcopenia was a significant, independent factor associated with frailty. A geriatric cross-sectional study from China showed that the likelihood of frailty was substantially higher among community-dwelling elders with osteosarcopenia [18]. A 4-year observational geriatric study from Japan reported that the occurrence of frailty significantly increased with an increased incidence of osteosarcopenia [17]. These findings indicate that osteosarcopenia and frailty are closely interrelated with each other. The present study is the first to demonstrate the relationship between osteosarcopenia and frailty in patients with CLD.

Muscle mass and bone mass are closely interrelated during growth [23]. Several factors (such as genetic predisposition, endocrine imbalance, nutritional states, and inflammation) can influence muscle tissues and bone metabolism. In the present study, we found that decreases in IGF-1 and BCAA were associated with osteosarcopenia and frailty. IGF-1, produced primarily in hepatocytes and some tissues (including bone and muscle), is involved in muscle protein synthesis and bone remodeling and the maintenance of bone mass and strength due to its stimulation of osteoblast differentiation and proliferation [23,24]. BCAAs, especially leucine, contribute to protein synthesis through the mammalian target of rapamycin pathway [25]. Given that the liver is a central organ for nutrient metabolism, LC is complicated by protein energy malnutrition and hyperammonemia and can lead to the consumption of BCAA by skeletal muscles for energy production and ammonia metabolism [26]. A reduced concentration of systemic BCAA is associated with sarcopenia in older adults and patients with LC [7,27]. Furthermore, frail participants showed lower daily consumption of protein and BCAA compared to both pre-frail and robust individuals [28]. Thus, our results are theoretically reasonable and support the notion that decreased levels of both IGF-1 and BCAA are associated with the development of osteosarcopenia and frailty in patients with CLD.

In the present study, both osteosarcopenia and frailty were closely associated with vertebral fractures and impaired physical performance in patients with CLD. Among community-dwelling elders, osteosarcopenia can greatly impair physical performance and balance, as well as increase the risk of falls and fractures, as compared to non-osteosarcopenia, sarcopenia, and osteoporosis alone [29,30]. Furthermore, vertebral fractures cause impairment in physical function and immobility [31], which can lead to a loss of muscle as well as bone mass and strength. In general, physical activity is essential for maintaining and improving BMD, muscle strength, and quality of the bone and muscle [32]. Multicomponent exercise interventions, combining resistance training with aerobic, balance and flexibility exercises, could be effective strategies in improving the muscle strength, gait speed, balance, and physical performance, and consequently reduce the fear of falling [33,34]. Pharmacologic treatment for osteosarcopenia remains to be developed; however, denosumab, a receptor activator of nuclear factor kappa-B ligand (RANKL) inhibitor, was recently reported to improve not only BMD but also muscle mass and strength in postmenopausal women with osteoporosis [35]. Large-scale studies should be conducted in the future to determine the optimal exercise and pharmacological treatment strategies for improving the functional and health-related outcomes in CLD patients with osteosarcopenia and/or frailty.

This cross-sectional study had some limitations. First, nutritional intake was not assessed. Second, the daily physical activity was not evaluated in detail. Third, we did not assess if osteosarcopenia preceded and/or initiated or facilitated frailty and vice versa, over a long-term observation period. Lastly, we did not evaluate if steroid therapy affected osteosarcopenia and frailty, although this study cohort included 7 patients who had been receiving prednisolone treatment for autoimmune hepatitis. A large-scale study that includes these assessments is needed to resolve the challenges related to osteosarcopenia and frailty in patients with CLD.

## 5. Conclusions

In the present study, we demonstrated that osteosarcopenia and frailty are prevalent and closely interrelated in patients with CLD. Specifically, patients with osteosarcopenia and/or frailty are susceptible to vertebral fractures that can lead to impairment in physical function. Therefore, comprehensive diagnostic assessments and appropriate treatment for these complications are crucial in patients with CLD.

## Figures and Tables

**Figure 1 jcm-09-02381-f001:**
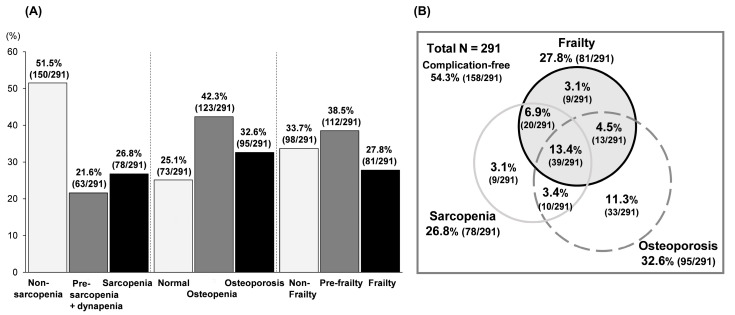
(**A**) Prevalence of sarcopenia/pre-sarcopenia and dynapenia, osteoporosis/osteopenia, and frailty/pre-frailty among patients with chronic liver disease. (**B**) Venn diagram showing the overlap between sarcopenia, osteoporosis, and frailty.

**Figure 2 jcm-09-02381-f002:**
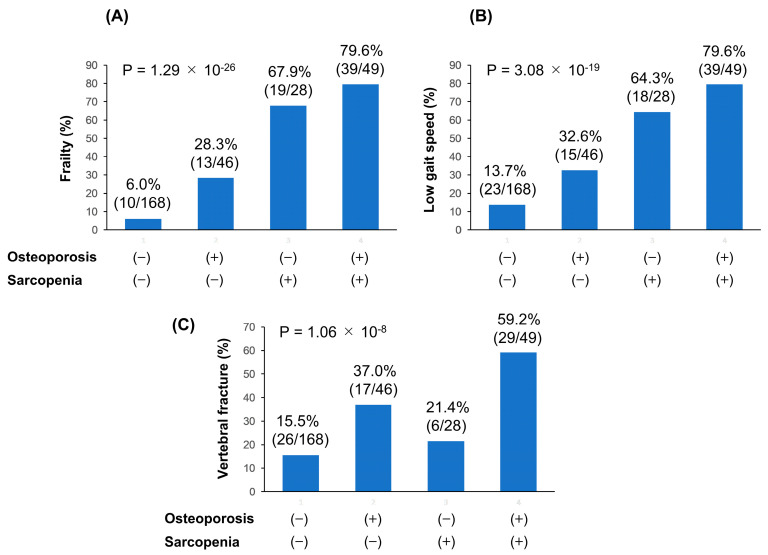
Comparison of clinical symptoms/events between the following four groups: (i) osteoporosis (−)/sarcopenia (−); (ii) osteoporosis (+)/sarcopenia (−); (iii) osteoporosis (−)/sarcopenia (+); (iv) osteoporosis (+)/sarcopenia (+) (i.e., osteosarcopenia) group. Among these four groups, the osteoporosis (+)/sarcopenia (+) group had the highest prevalence of (**A**) frailty (79.6% (39/49); *p* = 1.29 × 10^−26^, adjusted residual = |8.6|, Cramér’s V = 0.652); (**B**) low gait speed (79.6% (39/49); *p* = 3.08 × 10^−19^, adjusted residual = |7.7|, Cramér’s V = 0.554); (**C**) vertebral fracture (59.2% (29/49); *p* = 1.06 × 10^−8^, adjusted residual = |5.6|, Cramér’s V = 0.371).

**Figure 3 jcm-09-02381-f003:**
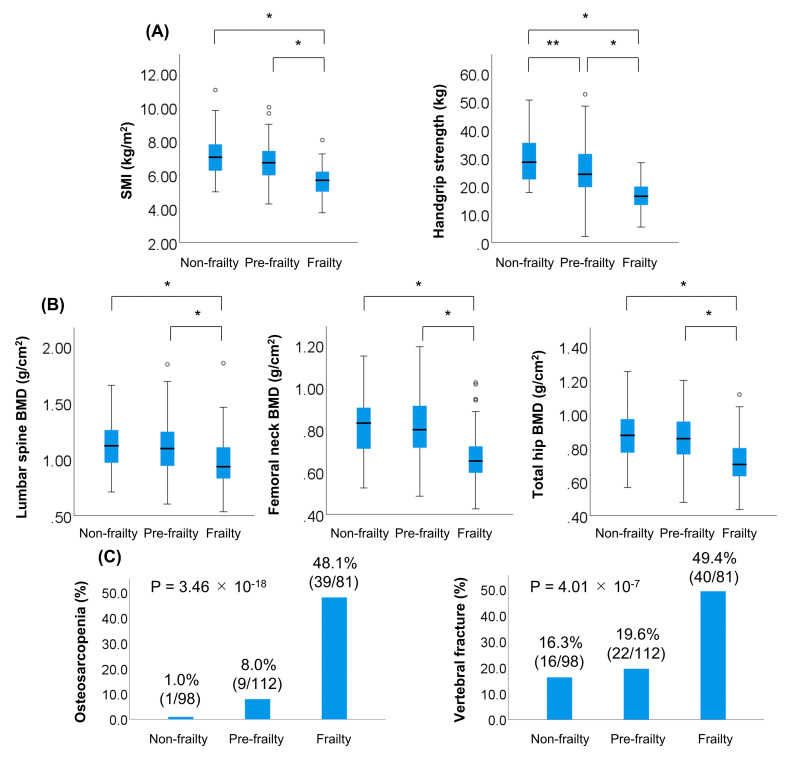
Comparison of clinical characteristics among the following three groups: (i) patients without both pre-frailty and frailty (non-frailty); (ii) patients with pre-frailty; (iii) patients with frailty. (**A**) The skeletal muscle mass index (SMI) and handgrip strength were significantly lower in the frail group than in the non-frail and pre-frail groups (*p* < 0.001 for both). (**B**) Bone mineral density (BMD) values of the lumbar spine, femoral neck, and total hip were significantly lower in the frail group than in the non-frail and pre-frail groups (*p* < 0.001 for all). (**C**) Among the three groups, the prevalence of osteosarcopenia was highest in the frail group (48.1% (39/81); *p* = 3.46 × 10^−18^, adjusted residual = |8.9|, Cramér’s V = 0.526), as was the prevalence of vertebral fracture (49.4% (40/81); *p* = 4.01 × 10^−7^, adjusted residual = |5.4|; Cramér’s V = 0.318). * *p* < 0.001, ** *p* < 0.05.

**Table 1 jcm-09-02381-t001:** Comparison of clinical characteristics between patients with and without osteosarcopenia.

Variable	All patients	Osteosarcopenia	Non-Osteosarcopenia	*p*-Value
Patients, *n* (%)	291	49 (16.8)	242 (83.2)	
Male, *n* (%)	137 (47.1)	16 (32.7)	121 (50.0)	0.027
Age (years)	70.0 (59.0–76.0)	76.0 (72.5–81.0)	68.0 (57.8–74.0)	<0.001
BMI (kg/m^2^)	23.1 (20.8–26.0)	20.4 (19.0–22.2)	23.7 (21.4–26.1)	<0.001
Liver cirrhosis, *n* (%)	151 (51.9)	31 (63.3)	120 (49.6)	0.081
Etiology				
HBV/HCV/alcohol/PBC/other, *n*	41/92/52/59/47	5/21/5/14/4	36/71/47/45/43	0.061
Total bilirubin (mg/dL)	0.7 (0.5–1.0)	0.6 (0.4–0.8)	0.8 (0.5–1.0)	0.006
Albumin (g/dL)	4.0 (3.7–4.3)	4.0 (3.6–4.3)	4.1 (3.7–4.3)	0.332
Prothrombin time INR	1.05 (0.97–1.15)	1.06 (0.97–1.11)	1.05 (0.97–1.15)	0.466
M2BPGi (C.O.I)	1.56 (0.86–3.58)	1.99 (1.37–2.96)	1.44 (0.78–3.75)	0.030
IGF-1 (ng/mL)	65 (47–90)	49 (41–64)	68 (50–95)	<0.001
Zinc (μg/dL)	68 (59–78)	66 (56–76)	68 (59–78)	0.378
BCAA (μmol/L)	405 (344–465)	333 (293–401)	417 (367–476)	<0.001
TRACP-5b (mU/dL)	415 (312–563)	463 (311–596)	410 (312–528)	0.273
P1NP (ng/mL)	49 (34–72)	46 (34–76)	49 (34–70)	0.714
PTH-intact (pg/mL)	47 (35–59)	51 (38–85)	44 (34–57)	0.010
SMI (kg/m^2^)				
All patients	6.48 (5.74–7.26)	5.12 (4.74–5.52)	6.75 (5.99–7.44)	<0.001
Male	7.19 (6.65–7.99)	5.93 (5.15–6.43)	7.34 (6.94–8.09)	<0.001
Female	5.88 (5.27–6.48)	4.94 (4.61–5.25)	6.04 (5.75–6.57)	<0.001
Handgrip strength (kg)				
All patients	23.4 (17.9–31.4)	15.1 (13.1–17.8)	24.8 (19.7–32.6)	<0.001
Male	31.0 (24.9–37.4)	21.8 (13.3–23.9)	32.4 (27.4–38.2)	<0.001
Female	18.9 (15.3–22.4)	14.9 (12.6–17.0)	21.0 (17.5–23.5)	<0.001
Lumbar spine BMD (g/cm^2^)	1.07 (0.90–1.22)	0.85 (0.75–0.95)	1.10 (0.95–1.24)	<0.001
Femoral neck BMD (g/cm^2^)	0.76 (0.67–0.89)	0.62 (0.56–0.65)	0.81 (0.71–0.90)	<0.001
Total hip BMD (g/cm^2^)	0.83 (0.71–0.94)	0.66 (0.58–0.71)	0.86 (0.76–0.96)	<0.001
Frailty, *n* (%)	81 (27.8)	39 (79.6)	42 (17.4)	<0.001
Low gait speed (m/s), *n* (%)	95 (32.6)	39 (79.6)	56 (23.1)	<0.001
Vertebral fracture, *n* (%)	78 (26.8)	29 (59.2)	49 (20.2)	<0.001

Values are shown as median (interquartile range) or number (percentage). Statistical analysis was performed using the chi-squared test or the Mann–Whitney U test, as appropriate. BCAA, branched-chain amino acids; BMD, bone mineral density; BMI, body mass index; HBV, hepatitis B virus; HCV, hepatitis C virus; IGF-1, insulin-like growth factor 1; INR, international normalized ratio; M2BPGi, Mac-2 binding protein glycosylation isomer; PBC, primary biliary cholangitis; P1NP, procollagen type N-terminal propeptide; PTH, parathyroid hormone; SMI, skeletal muscle mass index; TRACP-5b, tartrate-resistant acid phosphatase 5b.

**Table 2 jcm-09-02381-t002:** Factors associated with osteosarcopenia in patients with chronic liver disease.

Variable	Univariate		Multivariate	
	OR (95%CI)	*p*-Value	OR (95% CI)	*p*-Value
Male	0.485 (0.254–0.927)	0.029		
Age (years)	1.094 (1.053–1.136)	<0.001		
BMI (kg/m^2^)	0.767 (0.687–0.857)	<0.001	0.821 (0.726–0.929)	0.002
IGF-1 (ng/mL)	0.974 (0.961–0.987)	<0.001	0.980 (0.964–0.996)	0.014
BCAA (μmol/L)	0.993 (0.989–0.996)	<0.001		
PTH-intact (pg/mL)	1.019 (1.009–1.029)	<0.001	1.017 (1.005–1.030)	0.006
Frailty	15.545 (7.370–32.789)	<0.001	9.837 (4.199–23.043)	<0.001
Vertebral fracture	5.711 (2.981–10.943)	<0.001	3.306 (1.439–7.596)	0.005

BCAA, branched-chain amino acids; BMI, body mass index; CI, confidence interval; IGF-1, insulin-like growth factor 1; OR, odds ratio; PTH, parathyroid hormone.

**Table 3 jcm-09-02381-t003:** Comparison of clinical characteristics between patients with and without frailty.

Variable	Frailty	Non-Frailty and Pre-Frailty	*p*-Value
Patients, *n* (%)	81 (27.8)	210 (72.2)	
Male, *n* (%)	31 (38.3)	106 (50.5)	0.062
Age (years)	76.0 (69.5–80.0)	67.0 (56.0–73.0)	<0.001
BMI (kg/m^2^)	21.5 (19.8–24.5)	23.8 (21.3–26.2)	<0.001
Liver cirrhosis, *n* (%)	56 (69.1)	95 (45.2)	<0.001
Etiology			
HBV/HCV/alcohol/PBC/other, *n*	7/36/9/16/13	34/56/43/43/34	0.015
Total bilirubin (mg/dL)	0.7 (0.5–1.0)	0.7 (0.5–1.0)	0.140
Albumin (g/dL)	3.9 (3.4–4.3)	4.1 (3.8–4.4)	0.001
Prothrombin time INR	1.07 (0.99–1.17)	1.04 (0.97–1.14)	0.356
M2BPGi (C.O.I)	2.33 (1.29–4.70)	1.37 (0.72–2.71)	<0.001
IGF-1 (ng/mL)	54 (41–69)	72 (51–97)	<0.001
Zinc (μg/dL)	63 (51–75)	70 (60–78)	0.001
BCAA (μmol/L)	370 (308–412)	427 (370–485)	<0.001
TRACP-5b (mU/dL)	458 (323–592)	394 (311–528)	0.084
P1NP (ng/mL)	49 (33–79)	49 (36–67)	0.801
PTH-intact (pg/mL)	48 (33–68)	46 (35–57)	0.411
SMI (kg/m^2^)			
All patients	5.65 (4.97–6.24)	6.87 (6.05–7.59)	<0.001
Male	6.18 (5.65–6.94)	7.45 (7.03–8.16)	<0.001
Female	5.24 (4.81–5.75)	6.08 (5.71–6.57)	<0.001
Handgrip strength (kg)			
All patients	16.4 (13.2–20.2)	27.0 (21.2–34.3)	<0.001
Male	22.8 (18.3–24.7)	33.9 (29.4–38.8)	<0.001
Female	14.5 (12.3–17.0)	21.5 (18.5–23.9)	<0.001
Lumbar spine BMD (g/cm^2^)	0.93 (0.82–1.11)	1.11 (0.94–1.25)	<0.001
Femoral neck BMD (g/cm^2^)	0.65 (0.60–0.72)	0.82 (0.71–0.91)	<0.001
Total hip BMD (g/cm^2^)	0.70 (0.63–0.80)	0.86 (0.76–0.97)	<0.001
Low gait speed (m/s), *n* (%)	75 (92.6)	20 (9.5)	<0.001
Osteosarcopenia, *n* (%)	39 (48.1)	10 (4.8)	<0.001
Vertebral fracture, *n* (%)	40 (49.4)	38 (18.1)	<0.001

Values are shown as median (interquartile range) or number (percentage). Statistical analysis was performed using the chi-squared test or the Mann–Whitney U test, as appropriate. BCAA, branched-chain amino acids; BMD, bone mineral density; BMI, body mass index; HBV, hepatitis B virus; HCV, hepatitis C virus; IGF-1, insulin-like growth factor 1; INR, international normalized ratio; M2BPGi, Mac-2 binding protein glycosylation isomer; PBC, primary biliary cholangitis; P1NP, procollagen type *n*-terminal propeptide; PTH, parathyroid hormone; SMI, skeletal muscle mass index; TRACP-5b, tartrate-resistant acid phosphatase 5b.

**Table 4 jcm-09-02381-t004:** Factors associated with frailty in patients with chronic liver disease.

Variable	Univariate		Multivariate	
	OR (95%CI)	*p*-Value	OR (95%CI)	*p*-Value
Age (years)	1.089 (1.056–1.123)	<0.001	1.090 (1.050–1.130)	<0.001
BMI (kg/m^2^)	0.875 (0.811–0.943)	<0.001		
Liver cirrhosis	2.712 (1.574–4.672)	<0.001		
Albumin (g/dL)	0.375 (0.227–0.621)	<0.001		
M2BPGi (C.O.I)	1.124 (1.041–1.214)	0.003	1.149 (1.039–1.271)	0.007
IGF-1 (ng/mL)	0.980 (0.970–0.989)	<0.001		
Zinc (μg/dL)	0.974 (0.956–0.992)	0.006		
BCAA (μmol/L)	0.993 (0.989–0.996)	<0.001	0.994 (0.990–0.997)	0.001
PTH-intact (pg/mL)	1.009 (1.001–1.018)	0.006		
Osteosarcopenia	18.571 (8.596–40.121)	<0.001	10.069 (4.282–23.680)	<0.001
Vertebral fracture	4.416 (2.523–7.728)	<0.001		

BCAA, branched-chain amino acids; BMI, body mass index; CI, confidence interval; IGF-1, insulin-like growth factor 1; M2BPGi, Mac-2 binding protein glycosylation isomer; OR, odds ratio; PTH, parathyroid hormone.

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
