# Peer review of "Relationship between Osteosarcopenia and Frailty in Patients with Chronic Liver Disease"

_jcm, 2020, doi:10.3390/jcm9082381_

Round 1

Reviewer 1 Report

Authors have done a succint job of evaluating the prevalence of osteosarcopenia and frailty in patients with chronic liver disease. 

Comments: 

  • Patients with hypothyroidism, depression may present with symptoms of anhedonia overlapping with frailty. Did authors screen for other coexisting conditions which can act as a confounder?
  • Other medications including steroids in AIH can cause sarcopenia and osteopenia while causing mood symptoms. If not addressed or evaluated then should be added in the limitation section. 
  • Osteosarcopenia has been shown to be associated with frailty in various population and it would have been an interesting study to see if Osteosarcopenia in CLD patients is more prevalent compared to general population. 

Author Response

(1) Patients with hypothyroidism, depression may present with symptoms of anhedonia overlapping frailty. Did authors screen for other coexisting conditions which can act as a confounder?

Responses: We are deeply grateful for the reviewer’s critical comments. Hypothyroidism and depression can diminish physical and mental activities, and may be associated with frailty even in patients with chronic liver disease (CLD). A recent study suggested that sarcopenia was closely associated with depression in CLD patients (Nishikawa H, et al. J Clin Med 2019). When patients were strongly suspected to have hypothyroidism and depression, we examined them to diagnose the diseases. In this study, however, we did not screen all patients with these diseases in mind. In the future, we are willing to take into consideration various symptoms/diseases/conditions to clarify the association of osteosarcopenia and frailty with them in patients with CLD.

(2) Other medications including steroids in AIH cause sarcopenia and osteopenia while causing mood symptoms. If not addressed or evaluated then should be added in the limitation section.

Responses: We thank the reviewer for the above remarks. Certainly, treatment with prednisolone can cause osteosarcopenia. The present study included 10 patients with autoimmune hepatitis, of whom 7 had been receiving prednisolone treatment. We addressed this limitation on lines 354-356. Lastly, we did not evaluate if steroid therapy affected osteosarcopenia and frailty, although this study cohort included 7 patients who had been receiving prednisolone treatment for autoimmune hepatitis.

(3) Osteosarcopenia has been shown to be associated with frailty in various population and it would have been an interesting study to see if Osteosarcopenia in CLD patients is more prevalent compared to general population.

Responses: We deeply appreciate the reviewer’s encouraging comments. Our results suggest that patients with CLD are more susceptible to frailty as compared to the general population.

Reviewer 2 Report

Saeki and colleagues conducted an interesting study attempting to assess the existence of a relationship between osteosarcopenia and frailty in a large group of patients with chronic liver disease of various aetiologies. They proceeded to suggest an association between the two conditions and pointed out that their relation could lead to poor quality of life for these patients, as it enhances the risk of vertebral fractures and may impose restrictions in movement. Although I believe that the authors used sound statistical analyses to deal with their data and provide some interesting clinical results, I would like to comment on the following points. 1) As to the introduction part, it would be more interesting for the reader to have some more specific introductory comments on how the conditions are linked with chronic liver disease, for example if they are observed more often in virally-induced hepatitis as opposed to non-viral (NASH, ALD etc.) 2) As to the discussion part, I feel that the authors have not properly discussed their results, as they do not provide any explanation for any of the statistically significant associations (biochemical, genetic or cause of liver disease) they demonstrate in the results part. Therefore, it seems that the actual interplay between the underlying liver disease and the clinical conditions under investigation remains underexplored.

Author Response

Saeki and colleagues conducted an interesting study attempting to assess the existence of a relationship between osteosarcopenia and frailty in a large group of patients with chronic liver disease of various aetiologies. They proceeded to suggest an association between the two conditions and pointed out that their relation could lead to poor quality of life for these patients, as it enhances the risk of vertebral fractures and may impose restrictions in movement. Although I believe that the authors used sound statistical analyses to deal with their data and provide some interesting clinical results, I would like to comment on the following points.

(1) As to introduction part, it would be more interesting for the reader to have some more specific introductory comments on how the conditions are linked with chronic liver disease, for example if they are observed more often in virally-induced hepatitis as opposed to non-viral (NASH, ALD etc.).

Responses: We greatly appreciate the reviewer’s critical remarks. We fully understand and agree to the reviewer’s suggestion. Earlier, we also took notice of and examined differences in the prevalence of sarcopenia, osteoporosis, osteoporosis, and frailty among various liver diseases. However, baseline characteristics (e.g., age and gender) varied among different etiologies (attached file name for only review; etiology X); therefore, we were afraid that they would lead to misleading results and these situations might obscure the aim of this study. Here, we present the related backgrounds and results as follows:

HCV was the most frequent etiology in frail patients (p = 0.015; adjusted residual = |3.0|), whereas alcohol was the most frequent etiology in non-frail and pre-frail patients (adjusted residual = |2.2|) in this study. The median age of each etiology was 64.0 years for HBV, 75.0 years for HCV, 61.5 years for alcohol, 69.0 years for PBC and 72 years for others. As previously described [Ref. 7], HCV infection was highly prevalent in our area (Fuji city, Japan) in the beginning of the 1900s; therefore, patients with HCV-related CLD reach a more advanced age as compared to those with other liver diseases. In contrast, alcohol is the most common etiology among patients newly diagnosed with CLD. Accordingly, alcoholic patients were significantly younger than HCV patients. On our univariate and multivariate analysis, etiology was not significantly associated with frailty. Therefore, we could not conclude that patients with HCV-related CLD were more susceptible to frailty as compared to those with other liver diseases. As the next challenge suggested by the reviewer, a large-scale, multicenter study is needed to elucidate the relationship between frailty and etiology.

(2) As to the discussion part, I feel that the authors have not properly discussed their results, as they do not provide any explanation for any of the statistically significant association (biochemical, genetic or cause of liver disease) they demonstrate in the results part. Therefore, it seems that actual interplay between the underlying liver disease and clinical conditions under investigation remains underexplored.

Responses: We are deeply grateful for the reviewer’s critical comments. As suggested by the reviewer, we added a new paragraph to the revised text in the Discussion section (lines 317–333).

In the present study, we found that decreased IGF-1 and BCAA levels were associated with osteosarcopenia and frailty. IGF-1 is involved in muscle protein synthesis and bone remodeling and the maintenance of bone mass and strength due to its stimulation of osteoblast differentiation and proliferation. BCAA also contributes to protein synthesis through the mammalian target of rapamycin (mTOR) pathway. CLD with advanced disease stages is complicated by protein energy malnutrition and hyperammonemia and can lead to the consumption of BCAA by skeletal muscles for energy production and ammonia metabolism. Our results are theoretically reasonable and support the notion that decreased levels of both BCAA and IGF-1 are associated with the development of osteosarcopenia and frailty in patients with CLD.

Reviewer 3 Report

The current manuscript attempts to establish a relationship between Osteosarcopenia, Frailty and patients with CLD. The authors did a good job of presenting the material. The introduction does a good job of providing the context and describing the goal of the study, methods are adequately explained, results are well described and backed by detailed statistical analysis/comparisons. Finally, the authors discuss the results effectively and also provide conclusive arguments as well as describe some of the limitations to the study. Thus, overall, the content appears sufficient and although the relationships are solely established on the basis of statistical comparisons, however, the reported statistics appear adequate enough to support the arguments. 

The figures and graphs seem to be of low resolution. Specifically, Figure 1, 2, and 3 should be improved. These currently appear somewhat blurred and the text within these figures is difficult to read, which shouldn't be the case.  

Author Response

(1) The current manuscript attempts to establish a relationship between Osteosarcopenia, Frailty and patients with CLD. The authors did a good job of presenting the material. The introduction dose a good job of providing the context and describing the goal of the study, methods are adequately explained, results are well described and backed by detailed statistical analysis/comparisons. Finally, the authors discuss the results effectively and also provide conclusive arguments as well as describe some of the limitations to the study. Thus, overall, the content appears sufficient and although the relationships are solely established on the basis of statistical comparisons, however, the reported statistics appear adequate enough to support the arguments.

(2) The figures and graphs seemed to be of low resolution. Specifically, Figure 1, 2, and 3 should be improved. These currently appear somewhat blurred and the text within these figures is difficult to read, which shouldn't be the case.

Responses: We are deeply grateful for the reviewer’s encouraging comments. As suggested by the reviewer, we replaced Figure 1, 2, and 3 with better quality ones.

Reviewer 4 Report

Osteosarcopenia is a newly described syndrome that describes the co-existence of osteoporosis and sarcopenia, it is a global public health concern that will become increasingly relevant in the future. Frailty is a common clinical syndrome in older adults that characterized by a decline in multisystem functioning and physiologic reserve, with an increased risk for poor health outcomes including falls, disability, hospitalization, and mortality. The study conducted by Saeki et al was aimed to investigate the prevalence of osteosarcopenia and frailty complications, and the risk factors associated with osteosarcopenia and frailty in patients with chronic liver disease. This cross-sectional study included 291 CLD patients, with a median age of 70. Their data showed that osteosarcopenia and frailty are prevalent and closely interrelated in patients with CLD. The findings in this manuscript were innovative and offered new insights on understanding osteosarcopenia and frailty in patients with CLD. Also, this study indicated the importance of comprehensive diagnostic assessments and appropriate treatment for these complications in patients with CLD.

However, there is one issue need to be addressed for further consideration. The graphs of Figure 1, Figure 2, and Figure 3 are hardly readable, please replace them with better quality ones.

Author Response

Osteosarcopenia is a newly described syndrome that describes the co-existence of osteoporosis and sarcopenia, it is a global public health concern that will become increasingly relevant in the future. Frailty is a common clinical syndrome in older adults that characterized by a decline in multisystem functioning and physiologic reserve, with an increased risk for poor health outcomes including falls, disability, hospitalization, and mortality. The study conducted by Saeki et al was aimed to investigate the prevalence of osteosarcopenia and frailty complications, and the risk factors associated with osteosarcopenia and frailty in patients with chronic liver disease. This cross-sectional study included 291 CLD patients, with a median age of 70. Their data showed that osteosarcopenia and frailty are prevalent and closely interrelated in patients withCLD.

The findings in this manuscript were innovative and offered new insights on understanding osteosarcopenia and frailty in patients with CLD. Also, this study indicated the importance of comprehensive diagnostic assessments and appropriate treatment for these complications in patients with CLD. However, there is one issue need to be addressed for further consideration. The graphs of Figure 1, Figure 2, and Figure 3 are hardly readable, please replace them with better quality ones.

Responses: We are deeply grateful for the reviewer’s encouraging comments. We believe that comprehensive diagnostic assessments and appropriate treatment for osteosarcopenia and frailty are crucial in terms of health-related quality of life. As suggested by the reviewer, we replaced Figure 1, 2, and 3 with better quality ones.